# Epistemic Generative Adversarial Networks

## Abstract

Generative models, particularly Generative Adversarial Networks (GANs), often suffer from a lack of output diversity, frequently generating similar samples rather than a wide range of variations. This paper introduces a novel generalization of the GAN loss function based on Dempster-Shafer theory of evidence, applied to both the generator and discriminator. Additionally, we propose an architectural enhancement to the generator that enables it to predict a mass function for each image pixel. This modification allows the model to quantify uncertainty in its outputs and leverage this uncertainty to produce more diverse and representative generations. Experimental evidence shows that our approach not only improves generation variability but also provides a principled framework for modeling and interpreting uncertainty in generative processes.

## 1 Introduction

Generative models have achieved remarkable progress in synthesizing high-fidelity images, audio, and text, with Generative Adversarial Networks (GANs) standing out for their sample quality and training efficiency (Brock et al., 2018; Pei et al., 2021; Luo & Yang, 2024). These models have revolutionized various domains, from computer vision to natural language processing, demonstrating unprecedented capabilities in generating realistic synthetic data (Xu et al., 2018). Yet, despite numerous advances in objectives, architectures, and regularization techniques, lack of output diversity persists as a major challenge plaguing GANs and related generative approaches (Pei et al., 2021; Luo & Yang, 2024; Saad et al., 2023).

GANs often exhibit mode dropping and mode collapse, producing samples that look plausible but concentrate on a narrow subset of the target distribution (Rao et al., 2022; Pei et al., 2021). This phenomenon occurs when the generator fails to capture the complete range of diversity present in the training data, instead focusing on generating limited and repetitive variations of samples (Luo & Yang, 2024; Barsha & Eberle, 2024). Mode collapse arises from imbalances in the training dynamics between the generator and the discriminator (Rao et al., 2022). For example, if the discriminator is weak or learns too slowly, the generator can exploit this by converging to a narrow set of outputs that consistently fool the discriminator (Luo & Yang, 2024). The discriminator, seeing only limited patterns, gives the generator no incentive to explore other modes (Rao et al., 2022). Conversely, training instabilities can cause the generator to rapidly jump between modes, effectively collapsing to one mode at a time (Rao et al., 2022; Luo & Yang, 2024).

This imbalance erodes coverage, biases downstream analyses, and obscures failure modes that are crucial for trustworthy deployment (Rao et al., 2022; Cobbinah et al., 2025). The problem is particularly acute in conditional GANs, which tend to ignore the input noise vector and focus primarily on conditional information, further limiting diversity in generated outputs (Deja et al.; Thomas et al., 2024). In applications such as medical imaging, where diverse and clinically relevant outputs are essential, mode collapse poses critical challenges for practical deployment (Cobbinah et al., 2025; Saad et al., 2023).

Addressing the diversity problem requires not only better training dynamics but also principled approaches to quantify and leverage uncertainty in the generative process (Oberdiek et al., 2022; He et al., 2023). Traditional approaches to mitigate mode collapse have included architectural modifications, loss function adaptations, regularization techniques, and hybrid methods (Cobbinah et al., 2025). However, these solutions often come with computational costs or may reduce performance in scenarios where sample similarity is desired (Dubiński et al., 2022; Deja et al.). More fundamen-

tally, existing approaches typically lack a principled theoretical framework for understanding and modeling the uncertainty inherent in generative processes.

Uncertainty quantification in deep learning has gained significant attention, particularly for improving model reliability and trustworthiness (He et al., 2023; Oberdiek et al., 2022). In the context of generative models, uncertainty can arise from both data uncertainty (aleatoric) and model uncertainty (epistemic). Data uncertainty stems from inherent randomness and noise in the data, while model uncertainty arises from insufficient knowledge during training or prediction (He et al., 2023). Recent work has begun exploring uncertainty estimation in generative models, including approaches for diffusion models that estimate pixel-wise uncertainty to guide sampling processes (Vita & Belagiannis, 2024).

The Dempster-Shafer theory of evidence provides a robust mathematical framework for reasoning under uncertainty that generalizes classical probability theory (Dempster, 1967; Cuzzolin, 2023). Unlike traditional Bayesian approaches, Dempster-Shafer theory can handle incomplete and conflicting evidence by assigning belief masses to sets of propositions rather than single events (Dempster, 1967; Prieto et al., 2023). This framework has found applications across various domains, including sensor fusion, medical diagnosis, and collapse risk assessment (He et al., 2025; Wang et al., 2024; Wu et al., 2024). In computer vision, Dempster-Shafer theory has been successfully applied to tasks such as classification (Manchingal et al., 2025) and image segmentation (Scheuermann & Rosenhahn, 2010).

The theory's ability to model both uncertainty and ignorance makes it particularly well-suited for generative modeling applications, where we often face incomplete knowledge about the true data distribution and conflicting evidence from different modes (Prieto et al., 2023; He et al., 2025). Recent developments in evidential deep learning have begun incorporating belief theory into neural network architectures, though these approaches have faced challenges with zero-evidence regions that limit learning from certain training samples (Pandey & Yu, 2023).

Building on these foundations, this work introduces a novel approach that leverages Dempster-Shafer theory to address the diversity problem in GANs while providing principled uncertainty quantification. By extending the traditional GAN framework with evidence-theoretic principles and enabling region-wise uncertainty estimation, we aim to create generative models that can both capture greater diversity in their outputs and provide meaningful measures of confidence in their predictions. This approach represents a significant departure from conventional GAN training paradigms and offers new possibilities for creating more robust and interpretable generative models.

This is achieved through three key innovations: 1) modifying the discriminator to predict a belief function (Cuzzolin, 2010) rather than a traditional probability distribution for assessing image authenticity, 2) introducing architectural enhancements to the generator that enable region-wise uncertainty estimation via belief function prediction, and 3) developing a generalized GAN loss formulation that operates within the belief function framework.

**Contributions**. This paper makes the following contributions to the wider research in GANs:

1. We propose a unique Epistemic Generative Adversarial Network approach based on predicting belief functions in discriminators and generators. This leads to better generation quality and explainability of the model's generation.

2. We advance architectural enhancements to discriminator and generator architectures to enable them to work in a belief theoretical framework.

3. We set out a novel loss function to effectively train the proposed model in this framework while also introducing terms to balance stochasticity and diversity of the generator.

4. We show experimental results of our approach outperforming the standard model both in terms of image quality and diversity of generations on several datasets.

**Paper Outline.** This paper is structured as follows. Sec. 1 introduces the need for and concept of Epistemic Generative Adversarial Networks (E-GAN). Sec. 2 outlines the existing approaches and methods in this regard. In Sec. 3, we recall the notions of random sets, belief functions and mass functions. Sec. 4 details our E-GAN approach, loss and architecture. Finally, Sec. 5 provides empirical evidence supporting out approach and discusses the results. Conclusions are given in Sec. 6.

## 2 RELATED WORKS

This section provides a concise overview of existing methods that encourage diverse outputs using GANs in the literature. In general, these methods can be categorized into four groups: (1) Loss function modifications, (2) architectural modifications, and (3) training strategies and regularization.

Loss function directly impacts GAN training dynamics, thus influencing the generation diversity. Wasserstein GAN (WGAN) Arjovsky et al. (2017) is the most representative example. WGAN replaces the original Jensen-Shannon divergence with the Wasserstein-1 distance, improving mode coverage. Least Squares GAN (LSGAN) Mao et al. (2017) introduced a least squares loss to smooth the gradient and improve generation diversity. Following similar ideas, Hinge Loss GAN Lim & Ye (2017) penalizes bad generations only beyond a margin, leading to stabilized and diverse GAN training. Relativistic GAN (RGAN) introduced a relativistic discriminator formulation, stabilizing training by considering relative realism between real and generated samples, inherently balancing diversity and sample quality.

Architectural enhancement is an effective way to increase generator and discriminator complexity or introduce mechanisms to explicitly enhance diversity. InfoGAN Chen et al. (2016) implements the maximization of the mutual information between latent codes and generated samples. By doing so, the generator is forced to use the diversity of latent inputs and thus prevents mode collapse. Another common way is using multi-generator or multi-discriminator architectures. For example, MAD-GAN Ghosh et al. (2018) and MGAN Hoang et al. (2018) employ multiple generators, each learning different modes, explicitly dividing responsibility for diversity. While GMAN Durugkar et al. (2016) integrates multiple discriminators. PacGAN Lin et al. (2018), although following standard GAN structure, feeds multiple samples simultaneously to the discriminator, enabling straightforward detection of mode collapse through cross-sample correlation. BigGAN Brock et al. (2018) goes farther on architecture enhancements. It combined large-scale training (large batches, orthogonal regularization) to significantly enhance mode coverage and diversity.

Training strategies explicitly address the convergence stability issue, and indirectly prevent mode collapse by smoothing GAN learning dynamics. Early developed methods include Two-Time-Scale Update Rule (TTUR) Heusel et al. (2017), Unrolled GAN Metz et al. (2016), Adaptive Discriminator Augmentation (ADA) Karras et al. (2020), etc. Recent approaches emphasize more on the unification and explicit modelling of the latent-output relationship. For example, UniGAN Pan et al. (2022) introduced uniformity regularization within a normalizing flow GAN framework, explicitly targeting balanced mode coverage and reducing mode imbalance.

Overall, studies progress from initially addressing mode collapse through empirical heuristics and simple architectural modification, to rigorous theoretical improvements.

## 3 BACKGROUND

### 3.1 RANDOM SETS

Consider a meteorological station recording a 2D environmental vector, $\mathbf{e} = (T, H)$, where $T$ denotes temperature and $H$ denotes humidity, each modelled by a data probability distribution. Under normal operation the station reports exact values for both variables. Suppose, however, that the humidity sensor malfunctions and fails to return a numerical reading.

Rather than imputing an arbitrary humidity value, we represent the unknown component $H$ by the set $\mathcal{H}$ of all plausible humidity levels (e.g., $0\% \leq h \leq 100\%$). The underlying random process generating environmental conditions remains intact, but the observation becomes a set-valued random variable:

$$\mathbf{E} = (T, \mathcal{H}),$$

where $\mathcal{H}$ encapsulates the missing information. This model, in which random outcomes may be sets rather than points, defines a *random set*. Random sets extend classical probability by encoding uncertainty through collections of values instead of precise realizations, making them ideal for modeling imprecise or incomplete measurements (Matheron, 1975; Nguyen, 1978; Molchanov, 2005).

## 3.2 Belief functions & Mass functions

Random sets were formalized by Dempster (2008) and Shafer (1976a) as a framework for subjective belief, offering an alternative to Bayesian inference. When defined on finite domains—such as in classification—they are referred to as *belief functions*. Whereas a traditional probability mass function assigns normalized, nonnegative mass solely to individual outcomes $\theta \in \Theta$, a belief function distributes mass over subsets $A \subseteq \Theta$:

$$m(A) \geq 0 \quad \forall A \subseteq \Theta, \quad \sum_{A \subseteq \Theta} m(A) = 1. \tag{1}$$

The belief measure $Bel(A)$ quantifies the total mass supporting $A$ by summing masses of all its subsets. Conversely, one recovers $m$ via the Möbius inversion:

$$Bel(A) = \sum_{B \subseteq A} m(B), \qquad m(A) = \sum_{B \subseteq A} (-1)^{|A \setminus B|} Bel(B). \tag{2}$$

For instance, consider a diagnostic system classifying a patient into one of three disease categories $\Theta = \{flu, cold, allergy\}$. A belief assignment might be

$$m(\{flu\}) = 0.5, \quad m(\{allergy\}) = 0.2, \quad m(\{flu, cold\}) = 0.3.$$

Here, 50% of evidence supports *flu*, 20% supports *allergy*, and 30% supports the combined hypothesis $\{flu$ or $cold\}$ without distinguishing between them. The belief in $\{flu,cold\}$ is

$$Bel(\{flu, cold\}) = m(\{flu\}) + m(\{flu, cold\}) = 0.5 + 0.3 = 0.8,$$

indicating an 80% confidence that the patient has either flu or cold—capturing epistemic uncertainty rather than assigning hard probabilities to each disease (see Fig. 1). Belief functions thus generalize Bayesian probabilities by explicitly representing ignorance and partial knowledge, which is crucial when data are scarce or ambiguous (Shafer, 1976b; Bouckaert, 1995).

Belief functions have been applied in machine learning tasks such as classification (Cuzzolin, 2018) and regression (Gong & Cuzzolin, 2017; Cuzzolin & Frezza, 2000).

## 3.3 Continuous Belief Functions on Closed Intervals

Belief functions admit a natural extension to continuous domains by replacing discrete mass assignments with a mass *density* defined over the family of all closed intervals on the real line (Smets, 2005; Strat, 1984; Cuzzolin, 2020). Concretely, let $\mathcal{I} = \{[a, b] \mid a, b \in \mathbb{R}, a \leq b\}$ denote the set of closed intervals. A continuous mass density $m(a, b) \geq 0$ is specified on each interval $[a, b] \in \mathcal{I}$ and normalized so that

$$\int_{-\infty}^{+\infty} \int_{a}^{+\infty} m(a, b) \, \mathrm{d}b \, \mathrm{d}a = 1.$$

Each interval $[a, b]$, known as Borel interval, serves as a potential *focal element*, and because any interval contains infinitely many nested subintervals, the mass density covers an uncountable family of focal sets. Graphically, the support of $m(a, b)$ lies in the triangular region

$$\{(a, b) \in \mathbb{R}^2 \mid a \leq b\},$$

and one can visualize $m(a, b)$ as a surface over this half-plane (see Fig. 2).

Given an interval $[c, d]$, the *belief* in $[c, d]$ is obtained by integrating the mass density over all focal intervals contained in $[c, d]$:

$$Bel(X) = \int_{c}^{d} \int_{a}^{d} m(a, b) \, \mathrm{d}b \, \mathrm{d}a \tag{3}$$

# 4 Epistemic Generative Adversarial Networks

We tackle the enduring problem of insufficient variability in GAN-generated samples through a synergistic, twofold strategy.

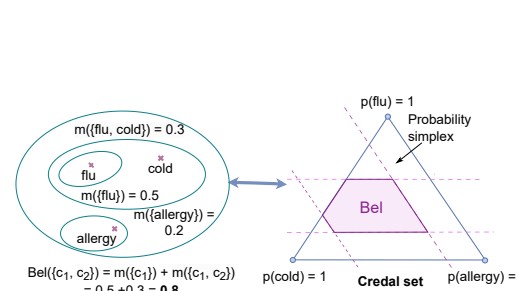

Figure 1: Belief functions and mass functions

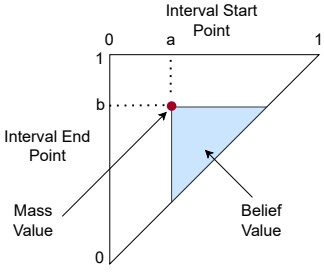

Figure 2: Representation of a belief function with Borel intervals

**Evidential loss generalization.** Drawing on the Dempster–Shafer theory of evidence, we derive a generalized adversarial loss that replaces conventional scalar discrimination with belief measures. In place of simple real/fake probabilities, both the discriminator and generator operate on belief functions over hypotheses of authenticity. Concretely, the discriminator assigns beliefs to the propositions "real" and "fake," along with composite sets when evidence is ambiguous. By explicitly modeling uncertainty in each adversarial decision, this loss compels the generator to sample from diverse regions of the latent space—reducing the tendency to collapse onto a few modes and enhancing the overall support of the learned distribution.

**Region-wise mass prediction.** To further enrich diversity and capture local ambiguity, we augment the generator's architecture with a dedicated *mass prediction head* at each region of pixels. Rather than outputting a single activation value, the network produces a normalized mass function over a small set of perceptual states. This region-wise uncertainty map not only highlights areas where the generator is least certain—such as object boundaries or texture transitions—but also encourages it to explore alternative plausible patterns. In practice, this mechanism yields images with richer detail variation and prevents repetitive textures that commonly arise under standard GAN training.

**Unified evidential framework.** By combining a belief-based loss with region-level mass prediction, our framework embeds epistemic uncertainty directly into both the training objective and the network's output representation. This unified evidential approach yields several benefits: it provides interpretable confidence maps for generated samples, and it offers fine-grained control over diversity via regularization. Overall, our contributions establish a principled methodology for designing GANs that are both more diverse in their outputs and more transparent in their decision-making processes.

## 4.1 ARCHITECTURE

To embed evidential reasoning within the GAN architecture, we introduce targeted modifications to both the discriminator and the generator. These enhancements enable each network not only to perform its primary task of distinguishing or synthesizing images but also to explicitly quantify and harness uncertainty throughout the adversarial process.

**Discriminator with belief outputs.** In conventional GANs, the discriminator concludes with a single scalar $D(\mathbf{x}) \in [0, 1]$, interpreted as the probability that input $\mathbf{x}$ is drawn from the real data distribution. While effective for binary discrimination, this approach disregards any measure of confidence behind the decision.

We instead replace this probability estimate with a *belief function*, whereby the discriminator predicts two normalized masses (see Fig. 3):

$$b_{\text{real}} = bel(\{\text{real}\}), \quad b_{\text{fake}} = bel(\{\text{fake}\}), \quad b_{\text{real}} + b_{\text{fake}} \leq 1.$$

These belief values $b_{\text{real}}, b_{\text{fake}} \in [0, 1]$ are produced via sigmoid activations on two output neurons. Aside from this dual-output layer, the remaining network—convolutional blocks, feature extractors,

and intermediate activations—remains unchanged. By encoding both the strength of support for "real" and for "fake," the discriminator conveys richer uncertainty information, enabling the generator to receive gradient signals that reflect not only correctness but also confidence levels.

**Generator with pixel-wise mass prediction.** Our generator is restructured into two sequential modules, designed to incorporate uncertainty into the synthesis pipeline (see Fig. 4):

1. *Mass Function Prediction Stage.* The first module takes as input a latent vector $z \sim p(z)$ and outputs, for each region $(i, j)$, a discrete mass function $m_{ij}$. Concretely, rather than yielding a deterministic activation value $x_{ij}$, the network produces parameters of a Dirichlet distribution $\mathrm{Dir}(\alpha_{ij})$ representing the mass function, whose support lies on the simplex. Sampling from this Dirichlet yields an interval, the width of which can represent the uncertainty at that region. A region here is a single activation when the intermediate output in generator is at scale lower than the final image.

2. *Interval Sampling and Image Construction Stage.* From each region's mass vector $m_{ij}$, we draw an interval. This intermediate map of intervals $I$, is then fed into the second module. The second module decodes this map $I$ into the final image $\hat{x}$ through upsampling and convolutional synthesis, effectively transforming region-level intervals into diverse visual structures.

**Dirichlet-based mass representation.** Directly modeling a continuous mass function over all possible real-valued intervals is intractable. We therefore represent each regions's mass function by a low-dimensional Dirichlet distribution $\mathrm{Dir}(\alpha)$ with three concentration parameters. This choice offers several advantages:

- It inherently enforces the simplex constraint $\sum_k m^k = 1$, matching the normalization requirement of mass functions.
- By varying the concentration parameters $\alpha$, the network can express both sharp (confident) and flat (uncertain) belief distributions.
- Mapping the 2D Dirichlet simplex to interval hypotheses provides a tractable proxy for continuous mass assignment.

By coupling an evidential loss at the discriminator with region-level Dirichlet mass predictions in the generator, our architecture explicitly captures epistemic uncertainty. This design not only enhances sample diversity but also yields interpretable uncertainty maps that can guide downstream tasks and analysis.

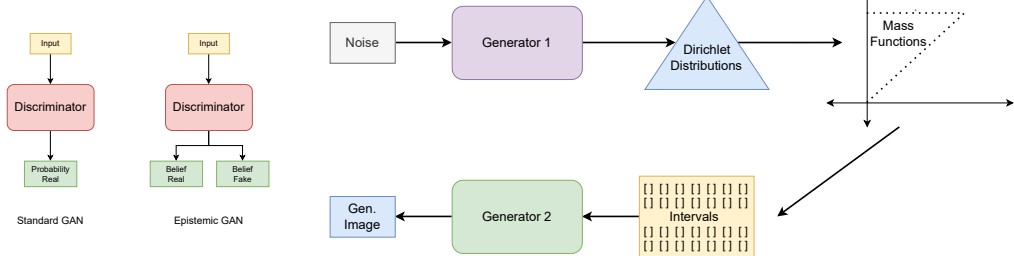

Figure 3: Discriminator architecture comparison.

Figure 4: Generator Architecture and flow for Epistemic GANs.

## 4.2 Loss Function

Because both networks now operate on belief masses rather than single-valued probabilities, we must redesign their adversarial objectives to incorporate evidential reasoning and maintain consistency with belief axioms. Below we present the redefined loss functions for the discriminator and generator.

**Discriminator loss.** Let $D$ denote the discriminator, $G$ the generator, and $z \sim p(z)$ a latent noise vector. Under our evidential framework, the discriminator outputs a belief pair

$$D(\mathbf{x}) = \big(b_{\text{real}}(\mathbf{x}),\, b_{\text{fake}}(\mathbf{x})\big),$$

where $b_{\text{real}}$ and $b_{\text{fake}}$ sum to at most one. To train $D$, we combine classical adversarial terms with regularizers that enforce coherent belief assignments:

$$
\begin{aligned}
L_D = &-\mathbb{E}_{\mathbf{x}\sim p_{\text{data}}}\Big[\log b_{\text{real}}(\mathbf{x}) + \log\big(1 - b_{\text{fake}}(\mathbf{x})\big)\Big] \\
&-\mathbb{E}_{z\sim p(z)}\Big[\log b_{\text{fake}}\big(G(z)\big) + \log\big(1 - b_{\text{real}}\big(G(z)\big)\big)\Big] \\
&+ \lambda\,\mathbb{E}_{\mathbf{x}\sim p_{\text{data}}}\big[\max\big(0,\, b_{\text{real}}(\mathbf{x}) + b_{\text{fake}}(\mathbf{x}) - 1\big)\big] \\
&+ \lambda\,\mathbb{E}_{z\sim p(z)}\big[\max\big(0,\, b_{\text{real}}\big(G(z)\big) + b_{\text{fake}}\big(G(z)\big) - 1\big)\big].
\end{aligned}
$$

The first two expectation terms drive $D$ to assign strong belief to real examples and reject fakes, analogous to the standard GAN loss. The subsequent regularizers, weighted by $\lambda > 0$, penalize any violation of the normalization constraint $b_{\text{real}} + b_{\text{fake}} \leq 1$, thereby preserving the integrity of the belief function.

**Generator loss.** The generator $G$ now produces, for each pixel $(i, j)$, a Dirichlet-parameter vector $\boldsymbol{\alpha}_{ij}$ that defines a discrete mass function over intensity intervals. To encourage both realism and diverse exploration of the latent space, we design $G$'s loss as

$$
\begin{aligned}
L_G = &-\mathbb{E}_{z\sim p(z)}\Big[\log b_{\text{real}}\big(G(z)\big) + \log\big(1 - b_{\text{fake}}\big(G(z)\big)\big)\Big] \\
&+ \beta\,\mathbb{E}_{i,j}\Big[\text{Var}\big[\text{Dir}\big(\boldsymbol{\alpha}_{ij}\big)\big]\Big] \\
&+ \gamma\,\mathbb{E}_{i,j}\big[w_{ij}\big].
\end{aligned}
$$

Here:

- The adversarial term mirrors the discriminator's objective, incentivizing $G$ to generate samples that receive high real-belief $b_{\text{real}}$ and low fake-belief $b_{\text{fake}}$.
- The variance term $\text{Var}[\text{Dir}(\boldsymbol{\alpha}_{ij})]$, weighted by $\beta > 0$, encourages the pixel-level Dirichlet distributions to spread mass across multiple categories, thus promoting mode exploration and richer image diversity.
- The mean interval width $w_{ij}$, weighted by $\gamma > 0$, is computed from the sampled intervals at each pixel; minimizing it drives the generator toward more precise, confident predictions where appropriate.

**Balancing diversity and precision.** By jointly maximizing Dirichlet variance and minimizing interval width, the generator navigates between two complementary objectives:

1. *Diversity promotion* via increased epistemic spread across intensity intervals, ensuring novel and varied patterns.
2. *Prediction confidence* through narrower interval selections, avoiding unrealistic or overly diffuse outputs.

This dual-objective scheme ensures that $G$ does not simply oscillate between noise and mode collapse but instead converges to a regime where generated images are both varied and coherent, with uncertainty maps that can be directly interpreted for downstream analysis.

## 5 EXPERIMENTS

### 5.1 IMPLEMENTATION

**Experimental datasets.** To comprehensively assess the performance of our evidential GAN framework, we conduct experiments across three distinct datasets that span different visual domains

and complexity levels. **CelebA** (Liu et al., 2015) serves as our primary benchmark for facial image generation, providing over 200,000 high-resolution celebrity portraits that have become a standard testbed for evaluating generative model quality and diversity. The dataset's rich variation in facial attributes, poses, and expressions makes it particularly suitable for assessing mode coverage and generation fidelity. **CIFAR-10** (Krizhevsky et al., 2009) offers a complementary evaluation setting with its 60,000 natural images distributed across ten object categories, enabling us to examine our method's ability to handle multi-class generation and maintain inter-class diversity. Finally, **Food-101** (Bossard et al., 2014) presents a more challenging scenario with 101 food categories, each containing 1,000 images, allowing us to evaluate scalability and performance on fine-grained visual distinctions within a specialized domain. This diverse collection of datasets ensures that our experimental validation spans multiple scales of complexity and visual characteristics.

**Implementation and training configuration.** We implement our evidential framework using the Deep Convolutional GAN (DCGAN) architecture (Radford et al., 2015) as our backbone, chosen for its established performance and widespread adoption in generative modeling research. Our experimental design focuses on a direct comparison with standard DCGAN to isolate the specific contributions of our evidential modifications, thereby providing a controlled evaluation of the proposed belief-based loss formulation and pixel-wise mass prediction mechanisms. We deliberately avoid comparisons with other state-of-the-art methods to maintain experimental clarity and ensure that observed improvements can be attributed directly to our theoretical contributions rather than architectural advances.

All hyperparameters governing our evidential terms are set uniformly: the belief regularization weight $\lambda = 1$, the variance promotion coefficient $\beta = 1$, and the interval precision weight $\gamma = 1$. This balanced parameterization ensures that no single component dominates the training dynamics. Training is conducted on NVIDIA A100 80GB GPUs with identical configurations across all experiments: 5 epochs of training, batch size of 128 samples, and the ADAM optimizer with standard learning rates ($\alpha = 0.0002, \beta_1 = 0.5, \beta_2 = 0.999$). To ensure reproducibility and fair comparison, both our evidential GAN and the baseline DCGAN are trained under identical computational constraints and random seed initialization. Image preprocessing follows standard practices with pixel values normalized to the range $[-1, 1]$ and all images resized to $64 \times 64$.

**Evaluation metrics.** To rigorously evaluate both the fidelity and the diversity of our evidential GAN outputs, we employ two complementary metrics: the Fréchet Inception Distance (FID) (Heusel et al., 2017) for image quality and the Vendi Score (Friedman & Dieng, 2022) for sample diversity.

FRÉCHET INCEPTION DISTANCE (FID). The Fréchet Inception Distance has become a standard metric for assessing the visual realism of images synthesized by generative models. FID operates by embedding both real and generated images into the feature space of a pretrained Inception network and then modeling each set of activations as a multivariate Gaussian distribution. Lower FID values indicate that the generated distribution more closely matches the real data distribution in feature space, reflecting higher sample fidelity and fewer visual artifacts.

VENDI SCORE. While FID effectively captures sample quality, it does not directly quantify how well a model covers the full support of the target distribution. To address this gap, we utilize the Vendi Score, a metric explicitly designed to measure diversity among generated samples. It is defined as the exponential of the Shannon entropy of the eigenvalues of a similarity matrix. Higher Vendi Scores thus reflect richer diversity, complementing FID's assessment of image fidelity.

## 5.2 RESULTS & ANALYSIS

Our experiments demonstrate that the proposed Epistemic GAN outperforms the standard GAN baseline in both image fidelity and generative diversity. Quantitatively, the Epistemic GAN achieves significantly lower Fréchet Inception Distance (FID) scores and higher Vendi Scores, indicating closer alignment with the real data distribution and broader mode coverage. Tab. 1 presents these numerical results across all datasets, while Fig. 5 offers representative visual comparisons.

Qualitatively, the Epistemic GAN generates a more diverse array of facial images. Although a few generated samples display minor artifacts—such as slight texture inconsistency or boundary irreg-

Table 1: Performance of Standard GAN and Epistemic GAN on Celeb-A dataset. Reference represents the Vendi score (diversity) of the training data.

| | CelebA | | Cifar-10 | | Food101 | |
|---|---|---|---|---|---|---|
| | FID ↓ | Vendi Score ↑ | FID↓ | Vendi Score ↑ | FID↓ | Vendi Score ↑ |
| Reference | - | 6.95 | - | 8.48 | - | 16.29 |
| Standard GAN | 18.5 | 5.70 | 25.9 | 4.25 | 33.76 | 12.78 |
| Epistemic GAN | **17.3** | **5.86** | **24.1** | **4.53** | **29.1** | **13.82** |

ularities—the overall visual quality remains superior to that of the standard GAN. These artifacts are infrequent and localized, suggesting that the evidential framework primarily enhances diversity without severely compromising sample coherence.

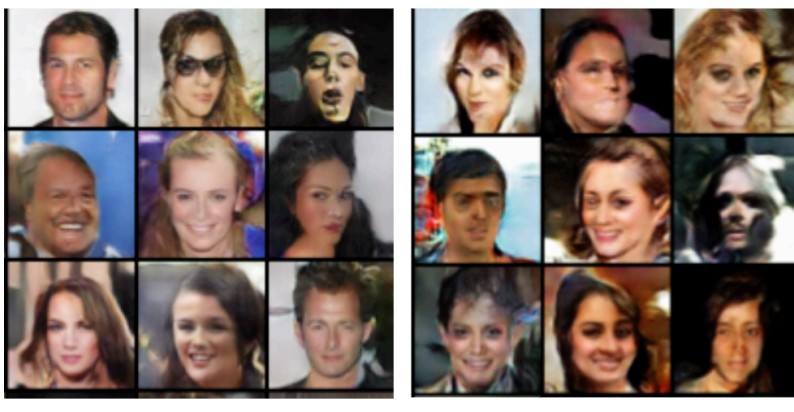

Standard GAN           Epistemic GAN

Figure 5: Generations for Standard GAN (left) and Epistemic GAN(right) for Celeb-A dataset.

## 6 CONCLUSION

In this work, we introduced the Epistemic GAN, a novel generative framework that integrates Dempster–Shafer evidence theory into both the adversarial loss and the network architectures. By replacing scalar probability outputs with belief functions in the discriminator and augmenting the generator to predict pixel-wise mass functions via Dirichlet distributions, our approach explicitly models epistemic uncertainty throughout the generative process. We derived generalized loss formulations that combine belief and plausibility terms with regularization constraints, and we designed a two-stage generator that samples interval hypotheses to encourage diverse and coherent outputs. Experiments on CelebA, CIFAR-10, and Food-101 demonstrated that the Epistemic GAN achieves lower Fréchet Inception Distances and higher Vendi Scores than the standard DCGAN baseline, indicating improvements in both image fidelity and mode coverage.

Our evidential framework opens several promising avenues for future research. First, extending the mass prediction mechanism to more complex uncertainty models—such as hierarchical belief structures or combining beliefs—could further enhance diversity and interpretability. Second, incorporating conditional evidential reasoning may facilitate controllable generation under ambiguous or missing labels. Finally, applying our approach to other generative paradigms (e.g., diffusion models or autoregressive architectures) could generalize the benefits of explicit uncertainty modeling across a wider range of synthesis tasks.

Overall, the Epistemic GAN offers a principled methodology for embedding uncertainty quantification into deep generative models, paving the way for more robust, interpretable, and diverse synthetic data generation.

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
