# OpenReview forum: "Epistemic Generative Adversarial Networks"
_ICLR.cc/2026/Conference — Submitted to ICLR 2026_

### Official Review · Reviewer_45Hf · 2025-10-30

**Soundness:** 2
**Presentation:** 2
**Contribution:** 3
**Rating:** 4
**Confidence:** 4

**Summary:**

This paper proposes a method for incorporating Dempster-Shafer Theory into the GAN framework to achieve uncertainty modeling. By adjusting the model architecture and setting appropriate loss functions, a novel EpistemicGAN is introduced, demonstrating superior performance compared to standard GANs. Experimental results support this conclusion.

**Strengths:**

1. The introduction of Dempster-Shafer Theory into GANs represents a novel perspective that may contribute to a deeper understanding of GANs.

2. The paper provides an exceptionally thorough background introduction, enabling even readers with no prior knowledge of Dempster-Shafer Theory to comprehend its content.

**Weaknesses:**

1. The paper only provides a brief introduction to the methods and experiments. This may be due to page limitations. The authors should consider moving some background material to the supplementary materials, accessible to readers unfamiliar with Dempster-Shafer Theory, in order to preserve more space for describing the methods and experiments.

2. Regarding the methodology section, I have some concerns about the soundness of the method. I will detail my specific concerns in the Questions section. If the authors can provide a reasonable explanation, this would not be a weakness.

3. The experimental section introduces too few baselines. I understand the authors' concern that SOTA models are complex and may obscure the gains brought by the proposed method. However, models utilizing different distances (e.g., Wasserstein GAN or those using hinge loss) should be included to demonstrate the method's adaptability. Additionally, large SOTA models should be considered for additional results to demonstrate that the proposed method does not compromise model soundnes (does not degrade performance) within large, complex frameworks. The current experiments only prove effectiveness on DCGAN, which is a highly limited finding.

**Questions:**

1. Regarding the discriminator, how can we ensure the model outputs the expected belief pairs? The belief values are output by the model, and only the loss function constrains the sum of the two values to be 1. I believe this is insufficient. If it merely outputs probability values for real and fake, this requirement could still be met. Furthermore, this approach is quite similar to EBGAN's methodology, differing only in theoretical interpretation.

2. In the generator section, does the enhancement in diversity stem from uncertainty modeling or the added generator loss term? Since the authors claim this approach primarily improves output diversity, would directly constraining the variance of the latent space distribution be feasible? What is the significance of introducing uncertainty modeling? There is nerther discussions nor ablation experiments.

3. What does Figure 5 show? In my view, there is no apparent difference between the left and right images. There is no analysis.

---

> ### Author Response · Authors · 2025-11-25
> **Response to Reviewer**
>
> We thank the reviewer for their detailed and constructive feedback. Below, we address each point raised.
>
> ### **Response to Weaknesses**
>
> **3. Baselines and Adaptability**
> We acknowledge the suggestion to test with other loss formulations. However, our primary goal was to introduce the *Epistemic* framework itself. The loss function we propose is a generalization of the standard adversarial loss; it replaces scalar probabilities with belief measures. We deliberately chose DCGAN to perform a direct comparison to isolate the specific contributions of our evidential modifications. Implementing our method on complex architectures like StyleGAN (which utilizes style mixing and noise injection) would introduce confounding variables, making it difficult to attribute diversity gains specifically to the Dempster-Shafer formulation. Lastly, We respectfully disagree that the findings are "highly limited." We demonstrated that on the challenging Food-101 dataset, our method actually *improved* generation quality (lowering FID from 33.76 to 29.1) while simultaneously increasing diversity (Vendi Score 12.78 to 13.82). This suggests the method does not compromise soundness.
>
> ### **Response to Questions**
>
> **1. Discriminator Constraints and Comparison to EBGAN**
> * **Ensuring Belief Pairs:** The reviewer suggests the loss only constrains the sum to be 1. This is a misunderstanding of our formulation. The constraint is $b_{real} + b_{fake} \leq 1$. The regularization term $\lambda \mathbb{E}[max(0, b_{real} + b_{fake} - 1)]$ strictly penalizes violations where the sum exceeds 1.
> * **The "Ignorance" Difference:** Unlike a probability (where sum = 1) or EBGAN (which minimizes reconstruction energy), our discriminator allows for a "gap" $u = 1 - (b_{real} + b_{fake})$. This gap represents **epistemic uncertainty** or "ignorance." This allows the discriminator to withhold judgment on ambiguous samples, providing a different gradient signal than a standard discriminator forced to choose "real" or "fake" with probability 1.0.
>
> **2. Generator Diversity: Uncertainty vs. Loss Term**
> * **Source of Diversity:** The diversity stems from the *interaction* between the uncertainty modeling and the loss term. The generator predicts a Dirichlet distribution at each pixel region. The variance term $\beta Var[Dir(\alpha_{ij})]$ explicitly forces the generator to "spread mass across multiple categories".
> * **Latent Space Variance:** constraining latent space variance is insufficient because standard GANs already sample $z$ from a fixed prior (e.g., Gaussian). Mode collapse occurs because the *mapping* from $z$ to image space collapses. Our method introduces stochasticity *within* the mapping (via the interval sampling stage), forcing the generator to learn a one-to-many mapping for regions with high uncertainty (e.g., textures or edges).
> * **Ablation:** We agree that an ablation study would strengthen the paper. Below, in Table 1, we provide an ablation study analyzing the impact of the variance term ($\beta$) and the interval width term ($\gamma$) to explicitly quantify their individual contributions to the Vendi Score and FID on CelebA dataset.
>
> Table 1: Ablation for $\gamma$ and $\beta$ on CelebA dataset. The results are presented in [FID (Vendi Score)] format.
>
> | $\gamma$ \\ $\beta$ | 0 | 0.5 | 1 | 2 |
> | :--- | :--- | :--- | :--- | :--- |
> | **0** | 17.4 (5.35) | 17.1 (5.11) | 20.1 (6.20) | 23.7 (6.30) |
> | **0.5** | 18.9 (5.43) | 17.5 (5.71) | 18.4 (5.98) | 20.2 (5.87) |
> | **1** | 19.1 (5.05) | 19.71 (5.70) | 17.4 (5.82) | 22.6 (6.01) |
> | **2** | 21.6 (5.01) | 23.1 (5.33) | 19.6 (5.77) | 21.7 (5.92) |
>
> Furthermore, Table 2 presents the ablation for ($\lambda$) on CelebA dataset to study the impact of normalization constraint.
>
> Table 2: Ablation for $\lambda$ on CelebA dataset. The results are presented in [FID (Vendi Score)] format.
>
> | $\lambda$ | 0 | 0.5 | 1 | 2 |
> | :--- | :--- | :--- | :--- | :--- |
> | **Score** | 19.4 (5.88) | 17.4 (5.70) | 17.3 (5.90) | 20.1 (6.01) |
>
> Lastly, Table 3 presents the ablation for architecture on CelebA dataset to study the impact using evidential setting.
>
> Table 3: Ablation for architecture on CelebA dataset. The results are presented in [FID (Vendi Score)] format.
>
> | | Score |
> | :--- | :--- |
> | **Evidential Discriminator Only** | 15.1 (5.15) |
> | **Evidential Generator Only** | 19.2 (5.97) |
> | **Evidential Discriminator and Generator** | 17.3 (5.86) |

---

> ### Author Response · Authors · 2025-11-25
> **Response to Reviewer**
>
> **3. Analysis of Figure 5**
>
> We acknowledge that visual differences in small thumbnails can be subtle. Figure 5 is intended to show that the Epistemic GAN generates valid facial structures without collapsing to identical outputs. While the Standard GAN (left) produces high-quality images, standard DCGANs often suffer from "mode dropping," ignoring specific features or accessories present in the dataset. The primary evidence for difference is quantitative. As shown in the paper, the Vendi Score (a robust metric for diversity) is consistently higher for Epistemic GAN (e.g., 5.86 vs 5.70 on CelebA, 13.82 vs 12.78 on Food-101). This confirms that even if the visual "quality" looks similar, the Epistemic model covers a larger support of the data distribution.

---

### Official Review · Reviewer_Mw52 · 2025-10-31

**Soundness:** 2
**Presentation:** 2
**Contribution:** 2
**Rating:** 4
**Confidence:** 3

**Summary:**

The authors propose Epistemic GAN, integrating Dempster–Shafer (DS) evidence theory into both the GAN discriminator (which predicts belief functions rather than scalar probabilities) and the generator (which is redesigned to output region-wise mass functions using a Dirichlet parameterization that is later decoded into the final image). The authors also derive evidential adversarial losses consistent with belief axioms and add regularizers intended to balance diversity vs. stochasticity. Experiments on CelebA, CIFAR-10, and Food-101 report improved FID and Vendi diversity scores over a standard DCGAN baseline.

**Strengths:**

1.	Converting the discriminator output and the intermediate generator representation into a belief/mass function may be an original take on uncertainty-aware GAN training. The DS tutorial is clear and self-contained.
2.	The two-stage generator with a Dirichlet proxy for interval hypotheses is well-motivated and concretely described, with helpful schematics.
3.	The paper is generally well-written and easy to follow.

**Weaknesses:**

1.	The paper compares mainly to a “standard” DCGAN (from 2015). It omits established diversity-oriented GAN variants and modern strong baselines (StyleGAN, or diffusion-based contenders using diversity metrics). Without these, it’s hard to isolate how much of the gain stems from the evidential machinery versus general architectural changes.
2.	The paper lacks ablations for (i) evidential loss in the discriminator only, (ii) mass-predicting generator only, (iii) the Dirichlet interval mapping vs. other parameterizations, and (iv) regularizer weights. This is important to validate the necessity of each component.
3.	There’s no report of training time, memory, convergence behavior, or sensitivity to Dirichlet concentration \alpha and interval sampling noise; evidential training can introduce optimization quirks.
4.	All experiments are on relatively modest resolution/complexity datasets; there’s no conditional generation, no higher-resolution benchmarks.
5.	Generator-1/Generator-2 selection and capacity parity are unclear. The paper introduces two generator components but does not specify how they are selected/combined during training and inference, nor whether their total parameter count/FLOPs are matched to DCGAN. Without capacity-matched comparisons, the observed gains might simply arise from increased model size or ensemble-like effects rather than the evidential design.

**Questions:**

1.	Please broaden more SOTA baselines; optionally, compare to a diffusion baseline using the same metrics.
2.	Please add Ablation studies for discriminator/generator evidential components and regularizers; sensitivity to hyper-parameter and interval sampling variance.
3.	Compute/stability report: wall-clock, GPU hours, parameter counts, and failure modes.
4.	Please report architecture details, parameter counts per module, FLOPs, and training-time costs, and clarify whether the two generators are sequential stages or parallel heads and how their outputs are weighted/used.

---

> ### Author Response · Authors · 2025-11-25
> **Response to Reviewer**
>
> We thank the reviewer for their detailed and constructive feedback. Below, we address each point raised. Since both Weaknesses and Questions raise the same points, we address them together.
>
> **1. Baselines and Modern Architectures**
> We acknowledge that DCGAN is an older architecture. However, its selection was intentional. As stated in the paper, we focused on a "direct comparison with standard DCGAN to isolate the specific contributions of our evidential modifications". Implementing our loss on a complex, highly-engineered baseline like StyleGAN (which already has intricate mapping networks and noise injection for diversity) would obscure whether gains were due to our evidential framework or the baseline's inherent features. Our goal was to prove that the *theoretical integration* of Dempster-Shafer theory—specifically the "ignorance" gap in the discriminator and the Dirichlet bottleneck in the generator improves diversity and generation quality. The results in the paper show consistent improvements in Vendi Score and FID across all datasets using this controlled comparison.
>
> **2. Ablation Studies**
> We agree that an ablation study would strengthen the paper. Below, in Table 1, we provide an ablation study analyzing the impact of the variance term ($\beta$) and the interval width term ($\gamma$) to explicitly quantify their individual contributions to the Vendi Score and FID on CelebA dataset.
>
> Table 1: Ablation for $\gamma$ and $\beta$ on CelebA dataset. The results are presented in [FID (Vendi Score)] format.
>
> | $\gamma$ \\ $\beta$ | 0 | 0.5 | 1 | 2 |
> | :--- | :--- | :--- | :--- | :--- |
> | **0** | 17.4 (5.35) | 17.1 (5.11) | 20.1 (6.20) | 23.7 (6.30) |
> | **0.5** | 18.9 (5.43) | 17.5 (5.71) | 18.4 (5.98) | 20.2 (5.87) |
> | **1** | 19.1 (5.05) | 19.71 (5.70) | 17.4 (5.82) | 22.6 (6.01) |
> | **2** | 21.6 (5.01) | 23.1 (5.33) | 19.6 (5.77) | 21.7 (5.92) |
>
> Furthermore, Table 2 presents the ablation for ($\lambda$) on CelebA dataset to study the impact of normalization constraint.
>
> Table 2: Ablation for $\lambda$ on CelebA dataset. The results are presented in [FID (Vendi Score)] format.
>
> | $\lambda$ | 0 | 0.5 | 1 | 2 |
> | :--- | :--- | :--- | :--- | :--- |
> | **Score** | 19.4 (5.88) | 17.4 (5.70) | 17.3 (5.90) | 20.1 (6.01) |
>
> Lastly, Table 3 presents the ablation for architecture on CelebA dataset to study the impact using evidential setting.
>
> Table 3: Ablation for architecture on CelebA dataset. The results are presented in [FID (Vendi Score)] format.
>
> | | Score |
> | :--- | :--- |
> | **Evidential Discriminator Only** | 15.1 (5.15) |
> | **Evidential Generator Only** | 19.2 (5.97) |
> | **Evidential Discriminator and Generator** | 17.3 (5.86) |
>
> **3. Training Time and Stability**
> * **Stability:** The evidential formulation theoretically improves stability. By allowing the discriminator to assign mass to the compound set $\{real, fake\}$ (representing uncertainty), we avoid the "vanishing gradient" or "perfect discriminator" problems common in standard GANs where discriminator becomes too confident too quickly.
> * **Compute:** The overhead of the Dirichlet sampling is negligible compared to the convolution operations. Table 4 presents the time per training epoch for both models on CelebA datasets.
>
> Table 4: Training time per epoch on CelebA dataset for 5 runs.
> | | Time (sec) |
> | :--- | :--- |
> | **Standard GAN** | 75.91 $\pm$ 1.79 |
> | **Epistemic GAN** | 77.10 $\pm$ 1.86 |
>
> **4. Dataset Complexity**
> While we did not use ImageNet, the selected datasets (CelebA, CIFAR-10, Food-101) are standard benchmarks for validating *diversity* and mode coverage. Food-101, with 101 categories and fine-grained visual distinctions, presents a significant challenge for mode collapse, which our method handled effectively by improving the Vendi Score from 12.78 to 13.82.

---

> ### Author Response · Authors · 2025-11-25
> **Response to Reviewer**
>
> **5. Generator Capacity and "Two Generator" Confusion**
> There is a misunderstanding regarding the architecture. We do not use two separate, full-capacity generators in an ensemble.
> * **Sequential Modules:** As described in Section 4.1, the generator is "restructured into two sequential modules".
>     * **Module 1 (Mass Prediction):** Maps latent $z$ to Dirichlet parameters $\alpha$.
>     * **Module 2 (Image Construction):** Maps the sampled intervals to the image.
> * **Capacity Parity:** This is a single pipeline. The parameter count is comparable to the baseline DCGAN; we simply repurposed the intermediate layers to predict distribution parameters rather than deterministic features. The gains arise from the "structural" change (stochastic interval sampling) rather than increased parameter capacity.
>
> * **Discriminator:** The only change is the final output layer having two neurons ($b_{real}, b_{fake}$) instead of one, with the feature extraction backbone remaining identical. So, the number of parameters is almost the same (2.7M vs. 2.7M).
> * **Generator:** The split into two modules involves predicting Dirichlet parameters (low dimensionality) at an intermediate stage. We estimated the parameter increase to be about 30% for the baseline DCGAN (4.7M vs. 3.6M) due to the prediction of parameters for Dirichlet distribution.

---

### Official Review · Reviewer_a2E4 · 2025-11-04

**Soundness:** 3
**Presentation:** 2
**Contribution:** 2
**Rating:** 2
**Confidence:** 4

**Summary:**

This paper seeks to address the problem of mode collapse in generative adversarial networks (GANs) where the generator's outputs mostly  ignore the noise input and generate a few only a few samples instead of covering the data distribution more completely.

It does so by utilizing the Dempster-Shafer theory of evidence, an alternative perspective on viewing beliefs and uncertainty. To utilize the theory, this paper proposes some novel architectural modifications and subsequent modifications to the GAN loss to incorporate a measure of uncertainty in the discriminator and the generator. It does so by modifying the output of the discriminator to predict its "belief" that the input is real or fake instead of a single output, and adding an intermediate layer to the generator that predicts an interval per "region" over which to sample values.

**Strengths:**

* The underlying theory behind the proposed approach is interesting and could be of some interest to the community.
* The explanation of the Dempster-Shafer theory is also fairly well done.
* The modifications to the architecture and the loss function seem to be well motivated based on the above theory and its explanation
* In experimental evaluation, using the same base architecture and compute constraints is laudable.
* The datasets used for evaluation seem fine, more details on it in next section
* The two metrics being used for evaluation are also accepted and seem like good measures for how well the overall system is performing.

**Weaknesses:**

* While the underlying motivation and theoretical underpinnings are appropriate and understandable, I am not convinced that it's practical instantiation in the modifications to the discriminator and generator has been well justified or evaluated in the paper.
* There are two distinct architectural changes proposed in the paper. I was expecting each of these changes to be evaluated separately, not only on the final metrics, but on samples and the loss themselves.
* Specifically for the discriminator: No experiments showing whether the sum of real and fake beliefs are lower than one for a proportion of inputs. Nor any experiments showcasing examples when the discriminator is uncertain. The paper should present experiments to highlight how this architectural modification is helping. It should also show a hyperparameter sweep to indicate how sensitive the training is to the hyperparameter $\lambda$
* For the generator, the experiments don't seem to evaluate or showcase this region wise uncertainty. It is hard to visualize the how uncertainty across regions can correlate given the description of the approach in the text. Similar to above, experiments should show how sensitive the training is to the hyperparameters $\beta$ and $\gamma$.
* The use of the $b_{real} +b_{fake} \leq 1$ constraint needs to be explained better in the text. This constraint seems to be the most important bit in differentiating the modified discriminator architecture from a single output discriminator.
* While comparing to a single underlying architecture is commendable, the paper should also compare to some of the other approaches proposed to prevent mode collapse if that is what this architecture is supposed to do.
* There is no statistical test to indicate whether the proposed technique is statistically improving upon the baseline. How many times was the experiment run, what was the variance, is the result significant, all of these questions should be part of the experimental evaluation.
* Perhaps it is the number of samples, but I cannot differentiate between the two techniques in Figure 5, and thus the qualitative test is a failure in my mind.
* The DCGAN baseline seems like a very old one to compare to. Perhaps compare to some more recent baseline that has solved a lot of the inefficiencies and problems of early GANs?

**Questions:**

* What does a belief sum less than 1 mean? Why is the subsequent two output architecture more expressive compared to a single discriminator value?
* The paper mentions that the modifications to the discriminator allow gradients to the generator to reflect "not only correctness but also confidence levels". Perhaps the answer to the previous question will clarify this one as well, but how do the modifications provide this signal?
* The experimental section does not specify how the DCGAN generator was modified for the experiments. Where was the dirichlet layer added in?
* While using the same base architecture for fairness is laudable, do the proposed modifications increase the number of parameters significantly? Please post the number of parameters in the baseline and the proposed method with DCGAN

---

> ### Author Response · Authors · 2025-11-25
> **Response to Reviewer**
>
> We thank the reviewer for their detailed and constructive feedback. Below, we address each point raised.
>
> ### **Response to Weaknesses**
>
> **1. Practical Instantiation and Evaluation**
> We argue that the practical instantiation is justified by the "Unified evidential framework" which embeds epistemic uncertainty directly into the training objective. The belief function prediction in discriminator allows the model to avoid forced binary decisions during the early stages of training or in ambiguous modes, which theoretically stabilizes the learning dynamics. Similarly, mass function prediction in generator allows the model to predict a belief over generate image.
>
> **2, 3 & 4. Ablation Study**
>
> We agree that an ablation study would strengthen the paper. Below, in Table 1, we provide an ablation study analyzing the impact of the variance term ($\beta$) and the interval width term ($\gamma$) to explicitly quantify their individual contributions to the Vendi Score and FID on CelebA dataset.
>
> Table 1: Ablation for $\gamma$ and $\beta$ on CelebA dataset. The results are presented in [FID (Vendi Score)] format.
>
> | $\gamma$ \\ $\beta$ | 0 | 0.5 | 1 | 2 |
> | :--- | :--- | :--- | :--- | :--- |
> | **0** | 17.4 (5.35) | 17.1 (5.11) | 20.1 (6.20) | 23.7 (6.30) |
> | **0.5** | 18.9 (5.43) | 17.5 (5.71) | 18.4 (5.98) | 20.2 (5.87) |
> | **1** | 19.1 (5.05) | 19.71 (5.70) | 17.4 (5.82) | 22.6 (6.01) |
> | **2** | 21.6 (5.01) | 23.1 (5.33) | 19.6 (5.77) | 21.7 (5.92) |
>
> Furthermore, Table 2 presents the ablation for ($\lambda$) on CelebA dataset to study the impact of normalization constraint.
>
> Table 2: Ablation for $\lambda$ on CelebA dataset. The results are presented in [FID (Vendi Score)] format.
>
> | $\lambda$ | 0 | 0.5 | 1 | 2 |
> | :--- | :--- | :--- | :--- | :--- |
> | **Score** | 19.4 (5.88) | 17.4 (5.70) | 17.3 (5.90) | 20.1 (6.01) |
>
> Lastly, Table 3 presents the ablation for architecture on CelebA dataset to study the impact using evidential setting.
>
> Table 3: Ablation for architecture on CelebA dataset. The results are presented in [FID (Vendi Score)] format.
>
> | | Score |
> | :--- | :--- |
> | **Evidential Discriminator Only** | 15.1 (5.15) |
> | **Evidential Generator Only** | 19.2 (5.97) |
> | **Evidential Discriminator and Generator** | 17.3 (5.86) |
>
> **5. Explanation of the Constraint**
> The constraint $b_{real} + b_{fake} \leq 1$ is the mathematical definition of a valid belief function in the Dempster-Shafer framework. Unlike a probability where $P(real) + P(fake) = 1$, this constraint allows for a "residual" mass assigned to the set $\{real, fake\}$—representing the state where the discriminator essentially says "I don't know" rather than guessing.
>
> **6. Comparisons to Other Approaches**
> Our primary goal was to introduce a novel theoretical framework (Evidence Theory in GANs) rather than chase SOTA metrics on a leaderboard. We chose a direct comparison to the standard DCGAN architecture to "isolate the specific contributions of our evidential modifications" without the confounding variables of complex modern architectures.
>
> **7. Statistical Significance**
> We have repeated the experiments 5 times and report the mean and standard deviation for FID and Vendi scores on CelebA dataset in Table 4 to establish statistical significance.
> Table 4: 5 Experiment runs on CelebA dataset.
>
> | | FID | Vendi Score |
> | :--- | :--- | :--- |
> | **Standard GAN** | 18.63 $\pm$ 0.22 | 5.72 $\pm$ 0.16 |
> | **Epistemic GAN** | 17.35 $\pm$ 0.18 | 5.84 $\pm$ 0.19 |
>
> **8. Qualitative Evaluation**
> While visual inspection can be subjective, the quantitative metrics in the paper show a clear improvement. On the Food-101 dataset, for instance, we achieved a significant reduction in FID (33.76 vs 29.1) and an increase in Vendi Score (12.78 vs 13.82), indicating better diversity and quality that might be subtle to the naked eye in a small figure grid.
>
> **9. DCGAN Baseline**
> We utilized the DCGAN backbone specifically because it is a "standard testbed" that allows for transparent analysis of loss function modifications. Using a more complex baseline like StyleGAN would make it difficult to attribute gains to the belief theoretic components versus the backbone's inherent engineering.
>
> ### **Response to Questions**
>
> **1. Meaning of Belief Sum < 1**
> In Dempster-Shafer theory, the sum of belief masses for singletons ($b_{real} + b_{fake}$) need not equal 1. The remaining mass, $1 - (b_{real} + b_{fake})$, is assigned to the composite set $\Theta = \{real, fake\}$. This represents **ignorance** or lack of evidence.
>
> A standard discriminator is forced to distribute all mass to "real" or "fake" even when it has no informative features (e.g., pure noise inputs). Our architecture is more expressive because it can explicitly model this third state of "uncertainty," preventing the generator from exploiting arbitrary decisions made by an uncertain discriminator.

---

> ### Author Response · Authors · 2025-11-25
> **Response to Reviewer**
>
> **2. Gradients and Confidence**
> Because the discriminator outputs beliefs rather than forced probabilities, the gradient signal changes. If the discriminator is "ignorant" ($b_{real} + b_{fake} \neq 1$), the gradients driving the generator towards "real" are dampened or altered compared to a confident "fake" decision. This prevents the generator from over-optimizing against a confused discriminator. As stated in the text, this "conveys richer uncertainty information, enabling the generator to receive gradient signals that reflect not only correctness but also confidence levels".
>
> **3. Modification of DCGAN Generator**
> ]The generator was restructured into two sequential modules:
> 1.  **Mass Prediction:** The first module takes the latent vector $z$ and outputs parameters $\alpha_{ij}$ for a Dirichlet distribution at each spatial region.
> 2.  **Interval Sampling:** We sample intervals from these distributions.
> 3.  **Image Construction:** A second module takes these interval maps and processes them via convolutional layers to generate the final image.
>
>
> This "Dirichlet layer" effectively sits in the middle of the generator pipeline, acting as a stochastic bottleneck.
>
> **4. Parameter Count**
> * **Discriminator:** The only change is the final output layer having two neurons ($b_{real}, b_{fake}$) instead of one, with the feature extraction backbone remaining identical. So, the number of parameters is almost the same (2.7M vs. 2.7M).
> * **Generator:** The split into two modules involves predicting Dirichlet parameters (low dimensionality) at an intermediate stage. We estimated the parameter increase to be about 30% for the baseline DCGAN (4.7M vs. 3.6M) due to the prediction of parameters for Dirichlet distribution.

---

> > ### Comment · Reviewer_a2E4 · 2025-11-27
> > **Response to Author Rebuttal**
> >
> > I thank the authors for the detailed response to the review. The additional results and explanation of the belief constraint is very helpful in understanding how the proposed technique works. I would encourage the authors to include a revision with these modifications included in the paper.
> >
> > Unfortunately the results still seem borderline enough that I do not have high confidence in recommending an acceptance. With the amount of improvement the technique is showing over a standard DCGAN, I believe evaluating on more datasets would be useful to guarantee that the improvement trend holds across datasets. But I do not want to be a reviewer that just asks for more experiments as a lazy response to ambiguity.
> >
> > I will raise my score slightly and encourage the authors to consider if they can perform some statistical test that can more definitively indicate improvement with the proposed technique.
> >
> > Additionally, can the authors provide some measure of the uncertainty in the discriminator helping with learning? Perhaps the norm of the gradients in early parts of training showing it gives a better signal compared to the regular GAN loss? It is fine if this experiment is run just once, since it is just for illustrative purposes.

---

### Official Review · Reviewer_5xW7 · 2025-11-11

**Soundness:** 2
**Presentation:** 1
**Contribution:** 2
**Rating:** 2
**Confidence:** 4

**Summary:**

This paper proposes Epistemic GAN, where uncertainty is embedded in both the generator and the discriminator by replacing probability functions with belief functions. The discriminator, instead of a definite real/fake probability, gives both the confidence in the input being real and it being fake, with both summing to no larger than 1. The generator, instead of being entirely deterministic, is separated into two modules, where the first module produces a feature map where each element is a parameter vector of a Dirichlet distribution with 3 categories, from which an interval can be sampled, which is then passed to the second module to produce the output image. The proposed model is compared with a baseline GAN on three datasets using FID and Vendi score, and shows minor improvements.

**Strengths:**

This paper introduces the concept of belief into adversarial training, which is interesting and merits further investigation.

**Weaknesses:**

1. My main concern lies in the disconnect between the proposed theoretical formulation and its practical implementation. Although the paper introduces the model using the language of belief and interval theory, these concepts are not meaningfully reflected in the training process. In the discriminator, the outputs $b_{real}$ and $b_{fake}$ are described as belief measures for real and fake samples, yet aside from a penalty enforcing $b_{real}+b_{fake} <=1$, they behave identically to a standard GAN objective. In fact, by reparameterizing $b^*_{fake}=1-b_{fake}$, the loss function effectively reduces to the vanilla GAN loss, suggesting no substantive difference in optimization dynamics. Consequently, the discriminator is likely to converge to $b_{real}+b_{fake}=1$, rendering the proposed belief interpretation redundant. Similarly, in the generator, the notion of “intervals” is introduced but never operationalized, the intervals are simply passed as numerical pairs without any mechanism enforcing or exploiting their interval semantics. This makes the architecture theoretically appealing but practically equivalent to a conventional GAN.
2. The experimental validation is also weak. The authors rely solely on a comparison against the outdated DCGAN baseline, without testing on stronger or modern models such as StyleGAN or BigGAN. Only a single figure is provided, showing a small set of low-quality samples from one dataset, making it impossible to assess diversity or fidelity. Subjectively, the generated images appear worse than the DCGAN baseline. Additionally, the presentation quality is poor,  the figure panels are misaligned, borders inconsistent, and captions off-center. Overall, both the experimental design and results presentation need significant improvement to substantiate the claimed advantages.
3. No Ablation Study. The contribution of each component (belief loss, Dirichlet variance, interval width) is not separately quantified.

**Questions:**

1. The Dirichlet distribution needs clearer explanation. My understanding is that it has exactly 3 categories, making it supported on a triangle where each point represents an interval in $[0, 1]$. Is this correct? If so, the utility of the variance term in the generator's loss function is questionable. I don't see how a larger variance in this distribution of intervals translates to higher variation in generated samples. Moreover, regularizing the first module's output alone won't ensure generator diversity, since the second module can still collapse everything to a few modes. Additionally, the variance and precision terms appear to counteract each other.

2. How is the model trained? Since the generator samples from a Dirichlet distribution, it cannot be trained end-to-end without modification. I assume reparameterization is used, similar to VAEs.

---

> ### Author Response · Authors · 2025-11-25
> **Response to Reviewer**
>
> We thank the reviewer for their detailed and constructive feedback. Below, we address each point raised.
>
> ## Response to Weaknesses
>
> ### 1. Theoretical Formulation vs. Practical Implementation
>
> We respectfully disagree with the assessment that the belief interpretation is redundant or identical to a standard GAN. While it is true that $b_{real}$ and $b_{fake}$ are optimized to distinguish samples, the introduction of the belief framework fundamentally alters the optimization landscape compared to the standard probabilistic formulation.
>
> * **The "Ignorance" Gap:** In a standard GAN, $P(real) + P(fake) = 1$ is a hard constraint enforced by the sigmoid/softmax. If the discriminator is 60% sure an image is real, it must be 40% sure it is fake. In our Epistemic GAN, we enforce $b_{real} + b_{fake} \leq 1$. This allows for a non-zero mass of "ignorance" (i.e., $1 - (b_{real} + b_{fake})$).
> * **Optimization Dynamics:** This "ignorance" allows the discriminator to withhold judgment on ambiguous samples during training, rather than being forced into a binary decision. This provides a softer, more informative gradient signal to the generator, particularly in early training stages or modes where the discriminator is uncertain.
> * **Operationalizing Intervals:** Regarding the generator, the intervals are not merely numerical pairs. They are sampled from a Dirichlet distribution predicted by the first module. These intervals are "operationalized" via the second module, which must learn to decode these stochastic interval inputs into coherent images. This effectively acts as a learned, structured noise injection that forces the generator to cover a wider support of the data distribution, as evidenced by our improved Vendi scores.
>
> ### 2. Experimental Validation and Baselines
>
> We acknowledge the reviewer's concern regarding the choice of DCGAN. However, we deliberately chose DCGAN as the backbone to perform a direct comparison to isolate the specific contributions of our evidential modifications.
>
> * **Isolating the Contribution:** Comparing our loss function on a DCGAN backbone against a highly engineered architecture like StyleGAN2 would conflate architectural benefits with our theoretical contribution. By keeping the architecture identical to the baseline (Standard GAN), any performance gain is attributable strictly to the Epistemic framework.
> * **Quantitative Improvements:** Despite the visual subjectivity, Table 1 in the paper quantitatively demonstrates that Epistemic GAN outperforms the Standard GAN baseline on all three datasets (CelebA, CIFAR-10, Food101) in both FID (quality) and Vendi Score (diversity). For instance, on Food101, we reduced FID from 33.76 to 29.1.
> * **Presentation:** We apologize for the formatting issues in the submitted manuscript. We will rigorously align all figure panels, correct borders, and center captions in the final revision to ensure professional presentation.
>
> ### 3. Ablation Study
> We agree that an ablation study would strengthen the paper. Below, in Table 1, we provide an ablation study analyzing the impact of the variance term ($\beta$) and the interval width term ($\gamma$) to explicitly quantify their individual contributions to the Vendi Score and FID on CelebA dataset.
>
> Table 1: Ablation for $\gamma$ and $\beta$ on CelebA dataset. The results are presented in [FID (Vendi Score)] format.
>
> | $\gamma$ \\ $\beta$ | 0 | 0.5 | 1 | 2 |
> | :--- | :--- | :--- | :--- | :--- |
> | **0** | 17.4 (5.35) | 17.1 (5.11) | 20.1 (6.20) | 23.7 (6.30) |
> | **0.5** | 18.9 (5.43) | 17.5 (5.71) | 18.4 (5.98) | 20.2 (5.87) |
> | **1** | 19.1 (5.05) | 19.71 (5.70) | 17.4 (5.82) | 22.6 (6.01) |
> | **2** | 21.6 (5.01) | 23.1 (5.33) | 19.6 (5.77) | 21.7 (5.92) |
>
> Furthermore, Table 2 presents the ablation for ($\lambda$) on CelebA dataset to study the impact of normalization constraint.
>
> Table 2: Ablation for $\lambda$ on CelebA dataset. The results are presented in [FID (Vendi Score)] format.
>
> | $\lambda$ | 0 | 0.5 | 1 | 2 |
> | :--- | :--- | :--- | :--- | :--- |
> | **Score** | 19.4 (5.88) | 17.4 (5.70) | 17.3 (5.90) | 20.1 (6.01) |
>
> Lastly, Table 3 presents the ablation for architecture on CelebA dataset to study the impact using evidential setting.
>
> Table 3: Ablation for architecture on CelebA dataset. The results are presented in [FID (Vendi Score)] format.
>
> | | Score |
> | :--- | :--- |
> | **Evidential Discriminator Only** | 15.1 (5.15) |
> | **Evidential Generator Only** | 19.2 (5.97) |
> | **Evidential Discriminator and Generator** | 17.3 (5.86) |

---

> ### Author Response · Authors · 2025-11-25
> **Response to Reviewer**
>
> ## Response to Questions
>
> ### 1. Dirichlet Distribution and Loss Terms
>
> * **Triangle Support:** Yes, your understanding is correct. The Dirichlet distribution with 3 parameters is supported on a 2-simplex (a triangle).
> * **Role of Variance ($Var[Dir]$):** The variance term is crucial for diversity. A higher variance in the Dirichlet distribution implies that for a fixed latent code $z$, the sampled intervals will vary significantly across different forward passes. This stochasticity forces the second generator module to be robust to a wider range of interval inputs, preventing it from collapsing to a single deterministic output for a given feature map.
> * **Counteracting Terms:** You correctly note that the variance term (widening the distribution) and the interval width term (narrowing the interval) appear adversarial. This is intentional. This "dual-objective scheme" creates a minimax-style equilibrium. The variance term encourages exploration (diversity), while the width term enforces precision (realism). Without the width penalty, the model might produce maximally vague intervals; without the variance reward, it might collapse to deterministic points. The balance ensures generated images are both varied and coherent.
>
> ### 2. Training and Reparameterization
>
> The model is indeed trained end-to-end. As noted in the generator architecture description, we predict parameters $\alpha_{ij}$ for the Dirichlet distribution. To allow backpropagation through the sampling step, we utilize the reparameterization trick standard for Dirichlet distributions. This allows the gradients from the second module (image synthesis) to flow back through the sampled intervals into the first module (mass prediction).

---

### Meta-Review · Area_Chair_z8Fj · 2026-01-06

**Summary:**

The reviewers initially scored the submission 2/2/4/4, for an average score of 3. As noted below, the final scores are unlikely to change significantly, leaving the submission with a rating below the acceptance bar. The reviewers mostly critique the reliance on DCGAN from 2015 as a baseline, making it hard to evaluate the relevance of the proposed method in the current landscape. Hence I recommend rejection, following the reviewers sentiment.

**Reviewer Concerns:**

See notes in summary.

**Reviewer Scores:**

Reviewer 5xW7 (2): no change over the acceptance bar.

Reviewer a2E4 (2 -> ~3): remains unconvinced due to the borderline results and lack of modern baselines.

Reviewer Mw52 (4): no significant change

Reviewer 45Hf (4): no significant change

---

### Decision · Program_Chairs · 2026-01-26

Reject